METHODS

# PoweREST: Statistical power estimation for spatial transcriptomics experiments to detect differentially expressed genes between two conditions

Lan Shui[1], Anirban Maitra[2], Ying Yuan[1], Ken Lau[3], Harsimran Kaur[3], Liang Li[1*], Ziyi Li[1*], **Translational and Basic Science Research in Early Lesions Research Consortia**

1 Department of Biostatistics, The University of Texas MD Anderson Cancer Center, Houston, Texas, United States of America, 2 Department of Translational Molecular Pathology, The University of Texas MD Anderson Cancer Center, Houston, Texas, United States of America, 3 Epithelial Biology Center, Vanderbilt University Medical Center, Nashville, Tennessee, United States of America

* zli16@mdanderson.org (ZL); lli15@mdanderson.org (LL)

**Data availability statement:** This study utilized publicly accessible datasets, including an intraductal papillary mucinous neoplasm

## Abstract

Recent advancements in spatial transcriptomics (ST) have significantly enhanced biological research in various domains. However, the high cost for current ST data generation techniques restricts the large-scale application of ST. Consequently, maximization of the use of available resources to achieve robust statistical power for ST data is a pressing need. One fundamental question in ST analysis is detection of differentially expressed genes (DEGs) under different conditions using ST data. Such DEG analyses are performed frequently, but their power calculations are rarely discussed in the literature. To address this gap, we developed PoweREST, a power estimation tool designed to support the power calculation for DEG detection with 10X Genomics Visium data. PoweREST enables power estimation both before any ST experiments and after preliminary data are collected, making it suitable for a wide variety of power analyses in ST studies. We also provide a user-friendly, program-free web application that allows users to interactively calculate and visualize study power along with relevant parameters.

## Author summary

Spatial transcriptomics technologies provide an unprecedented view of gene expression in tissues while preserving spatial context, enabling important discoveries in various biomedical fields, especially cancer research. However, the cost of profiling a single spatial transcriptomics slice typically ranges from $7,500 to $14,000, highlighting the importance of careful experimental design during the early planning stages. Oversampling can lead to unnecessary financial waste, while under-sampling risks insufficient

dataset (GSE233254) and a colorectal cancer dataset (DOI: 10.17605/OSF.IO/HFTQ2). All analysis code is available at https://github.com/lanshui98/PoweREST. The corresponding R package can be found on CRAN at https://cran.r-project.org/web/packages/PoweREST/index.html, and an interactive web application is accessible at https://lanshui.shinyapps.io/PoweREST/.

**Funding:** This work was funded in part by the Coordination and Data Management Center of the Translational and Basic Science Research in Early Lesions Program, which is supported by the National Cancer Institute grant U24CA274212 to LL and YY (https://reporter.nih.gov/search/IwLVMN6VeUy5C06mHVz5qw/project-details/10517004). The funders had no role in study design, data collection and analysis, decision to publish, or preparation of the manuscript.

**Competing interests:** The authors have declared that no competing interests exist.

statistical power, potentially resulting in a failure to detect true biological information. To address this challenge, we introduce a computational framework that estimates the statistical power for detecting differentially expressed genes in spatial transcriptomics experiments. Our method accounts for key factors when planning spatial transcriptomics studies, such as spatial information of gene expression within regions of interest, log-fold changes in gene expression between experimental conditions, gene detection rates, and number of slice replicates. In addition to a software package, we also provide a user-friendly, program-free web application that allows users to interactively calculate and visualize study power.

## 1. Introduction

Recent advancement in spatial transcriptomics (ST) enabled high-throughput measurements of transcriptomics while preserving spatial information about the tissue context [1]. Such advancement facilitated biological research in numerous fields of study, such as developmental biology, oncology, and neuroscience [2,3]. By incorporating transcriptomics and spatial data, ST data provide the opportunity to investigate human tissues from different research perspectives, such as identifying detailed tissue architecture, exploring domain-specific cell-cell interactions, and detecting differentially expressed genes (DEGs) in different regions [4–6]. Among these topics, DEG detection is a fundamental problem for disease mechanism investigation and biomarker discovery. After the tissue structures are identified using pathological or computational approaches, DEG detection helps explain the heterogeneity among different tissue regions and across different cell types. DEGs can serve as potential druggable targets for cancer treatment and diagnosis [7–10].

Although ST technology has been used in significantly advanced transcriptomic studies, the high cost of current ST profiling platforms limits the application of this technology in large-scale studies [11]. Several key experimental factors can affect signal generation in ST datasets, including the choice of tissue area, the number and sizes of the regions of interest (ROIs). Recent studies have provided insights into the effect of the number and sizes of ROIs on the statistical power of ST profiling, but their aim of power analysis is restricted to cell-type detection and cell–cell adjacency detection, rather than DEG detection [12,13]. Owing to the importance of DEG detection, the related power analysis was well developed for bulk RNA sequencing (RNA-seq) and single-cell RNA (scRNA)-seq experiments [5]. However, the literature contains little information on the power calculation in detecting DEGs using ST samples. Consequently, developing spatial power calculation tool for DEG detection has become a pressing need.

In transcriptomic studies, statistical power is usually influenced by parameters such as the desired error rate, the magnitude of the experimental effect of interest (effect size), and the sample size, which can be either the number of biological replicates or the number of cells or spots measured. In the case of detecting DEGs using bulk RNA-seq, the effect size is a gene's mean expression ratio (i.e., its fold change in expression) across two experimental conditions. Furthermore, because analysis of DEGs using bulk RNA-seq usually involves multiple genes, the problem of multiple comparisons must be addressed to reduce false positive discoveries [14]. For scRNA-seq studies in which cell-level information is available, DEG analysis can be further focused on comparisons under different conditions for a specific cell type or DEGs with differential expression across various cell types exposed to the same experimental conditions. Therefore, more factors in scRNA-seq studies can influence the study's power. Apart from effect size, the number of biological replicates and multiple testing methods, the number

of cells, and the proportion of cell types should also considered when determining a study's power [15,16].

The additional coordinate information available in ST data makes the power estimation for analysis of DEGs more complicated than that with bulk RNA-seq or scRNA-seq. Previous studies concentrated on estimating power for ST experiments aimed at detecting specific cell populations or identifying spatially variable gene expression patterns on tissue sections but not detecting DEGs [12,13]. To the best of our knowledge, only one recent spatial transcriptomics study has developed a dedicated power calculation strategy, using NanoString GeoMX data [17]. However, that study's method is not applicable to other popular ST platforms such as 10X Genomics Visium. Specifically, GeoMx supports free-form ROIs, which can be drawn to collect probes from any given region within the dimensions of 5-650 $\mu m$, whereas Visium measures gene expression in predetermined 55-$\mu m$-spot sizes with 100-$\mu m$ spaces between the centers of spots [18,19]. Thus, the power of GeoMx experiments is influenced by the ROI's shape and size, whereas the power of Visium experiments is determined by the number of spots. Additionally, the NanoString GeoMX study mentioned above [17] required prior Visium data for the simulation study. Furthermore, it provided no accessible tools for researchers inexperienced in coding, limiting its wide application.

To address the aforementioned challenges, we developed a **PoweR E**stimation Tool for **ST** Data, PoweREST (https://cran.r-project.org/web/packages/PoweREST/index.html), with an R Shiny app (https://lanshui.shinyapps.io/PoweREST/). PoweREST helps determine the optimal Visium ST experimental design for the detection of DEGs under two conditions. Unlike other power calculation methods for bulk RNA-seq or scRNA-seq, which assume gene expression follows Poisson or negative binomial distributions [20,21], PoweREST uses a nonparametric statistical power evaluation framework based on bootstrap to generate replicate ST datasets within ROIs. Moreover, our method employs the penalized spline (P-spline) [22] and XGBoost [23] under constraints that ensure a monotonic relationship of power with other parameters. Such monotone-respecting properties were not considered for the NanoString GeoMX power estimation method [17]. Our power estimation results across different ROIs and different tissue samples support PoweREST as a practical and reliable tool for power estimation for the detection of DEGs using ST data, helping researchers avoid both over-sampling and under-sampling during the early planning stages.

## 2. Materials and methods

### 2.1. PoweREST analytical framework

PoweREST evaluates the effect of experimental design on statistical power for ST datasets and helps select the optimal sample size for DEG detection. PoweREST uses a nonparametric framework and simulates different experimental scenarios based on a real Visium ST dataset to fully account for the complexity of ST data. PoweREST has four steps:

1. Bootstrap resampling of the spots within the ROI;
2. Differential expression (DE) analysis of the resampled spots;
3. Estimation of the statistical power using adjusted p-values for multiple testing;
4. Monotonic estimation of the statistical power surface using P-splines with XGBoost as a remedy.

A schematic overview of PoweREST's workflow with and without an available ST dataset is shown in Fig 1.

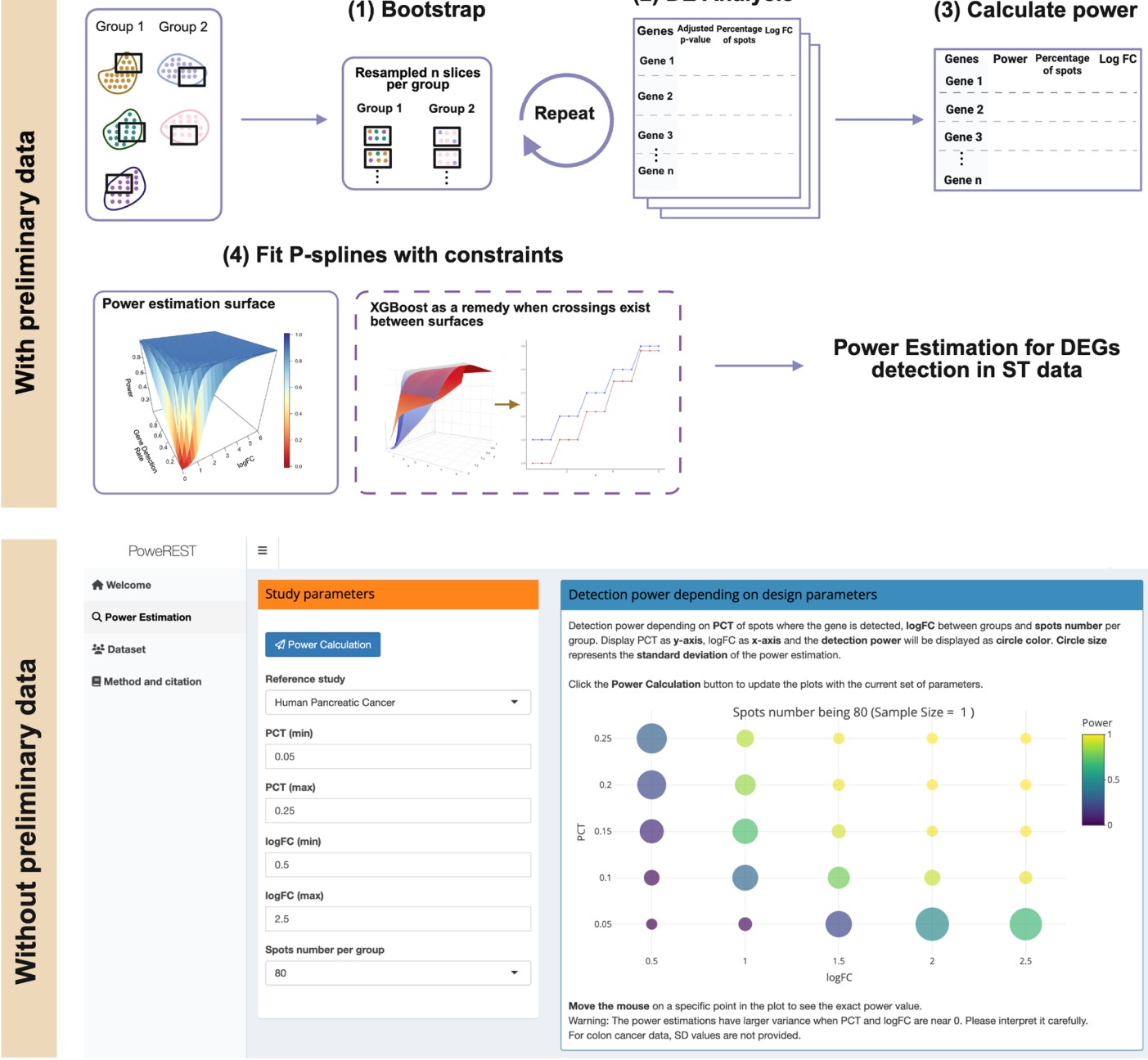

**Fig 1. Schema of the proposed PoweREST method.** When a preliminary cohort of ST data is available, PoweREST performs the power calculation based on bootstrap and P-splines fitting. When preliminary data are not available, an R Shiny app with power estimation results based on datasets from two cancer studies can be used. Created in BioRender. Shui, L. (2025) https://BioRender.com/injyq0j.

Users can apply PoweREST in two ways. First, when preliminary Visium ST data, which usually involves 2-3 samples on each arm, are available, users can employ the preliminary data as inputs and can use our R software package which is now available on CRAN to fit the problem-specific power surface. We also provide a tutorial website for the step-wise implementations of our software (https://lanshui98.github.io/powerest_tutorial/documentation/config.html). Second, users can directly set the parameters as the targeted effect sizes in the

PoweREST Shiny app (discussed in section 2.2) and obtain power calculation based on our models trained using publicly available datasets. The targeted parameters can be obtained from preliminary RNA-seq studies. For instance, consider a researcher investigating a novel treatment for pancreatic cancer. Based on preliminary RNA-seq analyses, the researcher aims to detect a set of immune-related genes with log-fold changes ranging from 0.5 to 3.6 between treated and untreated patient groups. To determine the required sample size, the researcher can utilize the power calculation software to estimate the number of ST spots necessary. Assuming an allocation of 50 spots per patient within the ROIs, our software can be used to compute the minimum number of patients required to achieve adequate statistical power for detecting the specified gene expression changes.

**2.1.1. Data resampling.** We assigned $C_1$ and $C_2$ as the true "population" ST datasets under condition 1 and 2, $c_1$ as the random ST samples under condition 1, and $c_2$ as the random ST samples under condition 2. We assumed selection of an average of $n$ spots in each slice's ROI across the two conditions, and such ROI selection are usually guided by H&E staining with the help from Pathologists. PoweREST creates ST specimen replicates within the ROI via bootstrap resampling [24,25]. Specifically, PoweREST randomly draws spot-level gene expression with a replacement from the sample data $c_1$ and $c_2$ to mimic the sampling process from the true "population" $C_1$ and $C_2$. We denoted the average detection rate of a gene across two conditions as $\pi_g$ and the average log-fold change in the expression between two conditions as $\beta_g$. Then, the power that we aimed to estimate was $Power_{ROI}(n, \pi_g, \beta_g, \alpha, N)$ with $\alpha$ being the desired adjusted p-value and $N$ being the target replicate (i.e., slice) number in each group. For simplicity in this report, we assumed the number of spots $n$ and the number of replicates $N$ were equal across the two conditions. However, both assumptions can be relaxed. That is, our method has the flexibility to accommodate imbalanced designs by allowing different numbers of replicates and different spot counts across experimental groups for power calculation. Nevertheless, we still require that the number of spots in the ROI is approximately consistent within each group. Since the number of spots $n$ is usually fixed in one experiment, which is determined by the selection of ROI, we used the number of tissue slices $N$, as the sample size and treated it as the influencing factor. Our bootstrap method implicitly incorporates the spatial information of ST data by computing test statistics using the spot-level gene expression within the same ROI. Rather than imposing a fixed spatial correlation structure, our nonparametric approach adapts to complex spatial relationships within the ROI, offering a more accurate representation of the power surface.

**2.1.2. Differential expression analysis.** After generating synthetic specimens based on preliminary Visium ST data through bootstrap resampling, our method implements the *FindMarkers* function from the Seurat software package for DE analysis [26]. Under default settings, this function analyzes the DE in two groups using the Wilcoxon rank sum test. Our method has the flexibility to perform DE analysis based on other statistical models. The detailed use can be found on our tutorial website. By applying the *FindMarkers* function to the resampled *in silico* replicates, for each gene, we can estimate the values of $\beta_g$, $\pi_g$ and the adjusted p-value $\alpha_g$, which is based on Bonferroni correction. According to the tutorial of the Seurat package [26], other correction methods are not recommended because *FindMarkers* prefilters genes, reducing the number of tests performed. These results were recorded and used for the power assessment in section 2.1.3.

**2.1.3. Power generation.** To estimate the statistical power, the previous two steps (bootstrap sampling and the DE analysis) are repeated enough times. By default, PoweREST repeated the two steps for 100 times and we assumed by repeating 100 times, the resampled *in silico* replicates within the ROI can represent the true population. This is based on the

assumption that within the ROI, the statistical power is not determined by the spatial context of spot-level gene expression. Thus, although bootstrap destroys the original spatial configuration, the power values are assumed to stay the same. Within every repetitions $i$, the genes with an adjusted p-value $\alpha_{gi}$ less than the desired adjusted p-value $\alpha$ are considered to be DEGs. The power of DEG detection is calculated using Eq 1.

$$Power_{ROI} = \frac{\sum_{i=1}^{100} I(\alpha_{gi} < \alpha)}{100}.$$  (1)

**2.1.4. P-splines fitting.**   After the previous three steps, power values for DEG detection are derived under different combinations of values of $N, \pi_g$ and $\beta_g$. To estimate the power under a new combination of $\pi_g$ and $\beta_g$ values under a sample size $N$, PoweREST uses 2D P-splines with monotonic constraints to fit a power surface.

When the sample size $N$ is fixed, one can assume that power value increases as $\pi_g$ or $|\beta_g|$ increases. We can also infer such monotonic relationship from the mathematical formula which is included in the S1 Appendix. Because of this relationship, unconstrained nonparametric models may be too flexible and give implausible or uninterpretable results. Our method uses shape-constrained additive models [27,28] to fit the power surface while preserving the monotonicity between power and both the $\pi_g$ and $|\beta_g|$) parameters using the 2D smooth function $m(\pi_g, |\beta_g|)$. Specifically, for a univariate smooth spline function $f(x) = \sum_{j=1}^{q} \gamma_j b_j(x)$, where $q$ is the number of basis functions, $b_j$'s are B-splines basis functions, and $\gamma_j$'s are unknown coefficients. To smooth $f$ while also ensuring a monotonic relationship between $f$ and $x$, a smoothing penalty and a shape constraint are imposed upon $\gamma_j$'s. We use quadratic splines and apply the Newton-Raphson method to maximize the penalized likelihood for estimation of the $\gamma_j$'s. The estimations are robust to the choice of $q$ when shape constraints are employed [29]. The statistical expression and derivation of the bivariate P-splines $m$ under double penalties and double monotonicity can be found in S2 Appendix.

**2.1.5. XGBoost as an ad-hoc approach for failure to maintain the monotonic relationship between power and sample size.**   P-splines with 2D monotonic constraints ensure the monotone relationships between power and both the $\pi_g$ and $|\beta_g|$, but the monotonic relationship between power and sample size $N$ is not ensured. Currently, a robust software for P-splines under 3D monotonic constraints is not readily available. In practice, we found that the estimated power values keep their monotonicity relationship with sample size when $\pi_g$ and $\beta_g$ are large, but such monotonic relationships dissolve in some cases when both $\pi_g$ and $\beta_g$ are close to 0. To address this deterioration in relationship, we propose employing XGBoost [23] to impose 3D monotonic constraints on $\pi_g, |\beta_g|$ and $N$ to estimating power values when $\pi_g$ and $\beta_g$ are small.

XGBoost solves the fitting problem using decision tree ensembles, which sum up the decision values of multiple trees to make a final decision. The tree structure is trained through an additive strategy: in every step, fix the learned tree and select the new leaf that optimizes the current objective. The monotonic constraints are achieved using the approach that at every step, abandon a candidate split if it causes a nonmonotonic relationship. However, because the algorithm essentially treats the fitting problem as a decision-making procedure, it visually fits a step function rather than a smoothing curve. Therefore, using XGBoost to fit the entire power surface can result in a crude fit. Thus, we recommend that users begin with P-spline fitting and carefully examine the resulting fits. When power surfaces for different sample sizes intersect, XGBoost can be applied specifically for regions in which these crossings occur. Thus, here we proposed XGBoost as an ad-hoc approach for failure to maintain

the monotonic relationship between power and sample size. We illustrated how to inspect the resulting fits from P-spline and how XGBoost could deal with it in section 3.

## 2.2. Implementation of R software package and R Shiny app

We implemented the proposed methods in an open-source R package named PoweREST. A tutorial for using PowerREST is included on the GitHub pages (https://github.com/lanshui98/PoweREST) and CRAN page (https://cran.r-project.org/web/packages/PoweREST/index.html), and contains detailed instructions for and examples of using the package and interpreting the results. To facilitate the application of PoweREST by users who are unfamiliar with R coding, we also created an online, interactive, program-free web application using R Shiny. As shown in Fig 1, users can select the tissue type with targeted parameter values for the ST experiments, and the study power can be generated by clicking the "calculate" button in the web page.

All the analyses in this manuscript were performed in R version 4.3.1.

# 3. Results

## 3.1. Power surface estimation with human intraductal papillary mucinous neoplasms data

The first dataset we examined was a publicly available 10X Genomics Visium dataset (GSE233254) on human intraductal papillary mucinous neoplasm (IPMN) tissues [10]. The dataset contains 13 specimens with 12,685 spots and up to 8,000 detected genes. Of the 13 specimens, 6 are classified in the high-risk (HR) IPMN category, and 7 are classified in the low-grade (LG) IPMN category. IPMNs are bona fide precursor lesions of pancreatic ductal adenocarcinoma. Clinically, HR lesions with or without an associated invasive cancer require surgical resection [10]. To reveal area-specific DEGs between two groups, the datasets provide the annotated spots that overlap with the neoplastic epithelium (epilesional, n=755); the immediately adjacent microenvironment, which corresponds to two layers of spots ($\sim$200$\mu$m) surrounding the lining epithelium (juxtalesional, n=1,142); and an additional two layers of spots located further distal to the juxtalesional region (perilesional, n=1,030) based on hematoxylin and eosin staining. Fig 2A and 2B show two representative slices of such histologically direct spot annotations. Within each of the three regions, we resampled the spots to create simulated data within the region and performed the proposed power estimation method.

**3.1.1. Power of DE analysis in perilesional areas.** Of the 1,030 perilesional spots, 540 were identified in HR samples, and 490 were identified in LR samples. We resampled 240,320,...,720,800 spots 100 times from the HR samples and the same number of spots from the LG samples to mimic the regional specimens of 3,4,...,9,10 replicates under each condition. We resampled the spots at an increment of 80 because we observed roughly 80 spots in each region per specimen. Using PoweREST, the power surfaces were fitted smoothly for different combinations of logFC and gene detection rate while maintaining the monotonic relationships between power and both logFC and detection rate. The fitted surfaces for the three selected replicate values are shown in Fig 2C. Although the power surfaces were estimated separately for each replicate value, the monotonic relationship between power and number of replicates per group remained (Fig 2D). We further validated the power estimation results by randomly selecting a subset of slices from the dataset and comparing observed DE results to the predicted power. With 6 slices per group, our proposed method suggests

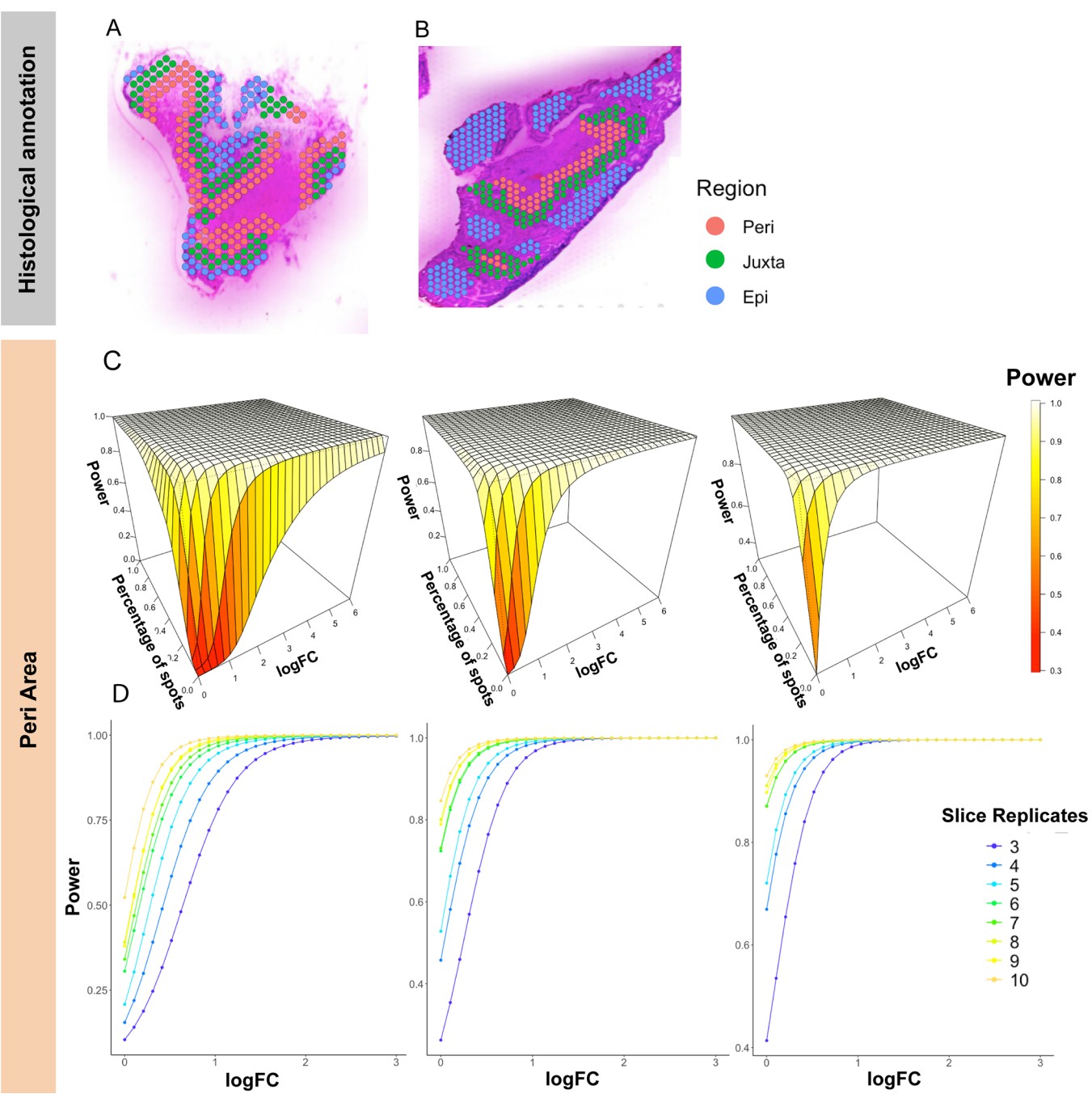

**Fig 2. Power estimation based on the IPMN dataset.** (A and B) Histologically annotated epilesional, juxtalesional, and perilesional spots are shown in an (A) LG sample and (B) HR sample. (C) The fitted power surfaces for sample sizes of 6, 8, and 10 per group within perilesional ('Peri') areas. The power was fitted under the constraints so that it monotonically increases with the percentage of spots where the gene is detected and the logFC in gene expression. (D) The relationships between the power and logFC when the percentage of spots detecting the gene equals 0.1, 0.2, and 0.3. Although we did not force the monotonic relationship between sample size and power, their relationship was still monotonic in the fitted results.

a statistical power of 0.92 for detecting a DE gene with a detection rate of 0.1 and an absolute log-fold change of 0.6. Our validation results (S1 Fig and S1 Table) confirm this power assessment.

**3.1.2. Power of DE analysis in juxtalesional and epilesional areas.** To validate the proposed method across different ROIs in the same tissue sample, we repeated the analysis in the other two annotated areas of IPMN tissues. For the juxtalesional areas, of the 1,142 spots, 568 spots were identified in HR samples, and 574 were identified in LG samples. Among the 755 spots in epilesional areas, 441 spots were identified in HR samples, and 314 spots were identified in LG samples. To obtain comparable results with perilesional areas, the spots were still resampled in increments of 80. Again, PoweREST estimated the power under different values of parameters using P-splines. The power results for juxtalesional and epilesional areas, which were similar to those for the perilesional area, are presented in Figs 3A, 3B and S2. The relative differences in the power results between the perilesional and juxtalesional and epilesional areas were calculated (S3 Fig). We found that the differences were minimal and only existed in regions where both the logFC and percentage of expressed spots were close to 0. The relative difference in the fitted power values between the juxtalesional and perilesional areas ranged from 1.8e-10 to 1.2, whereas the difference between the epilesional and perilesional areas ranged from 9.5e-09 to 7.4 with 7.4 being the relative difference when the logFC and percentage of spots with gene expression were 0 and the fitted power values were 0.3 and

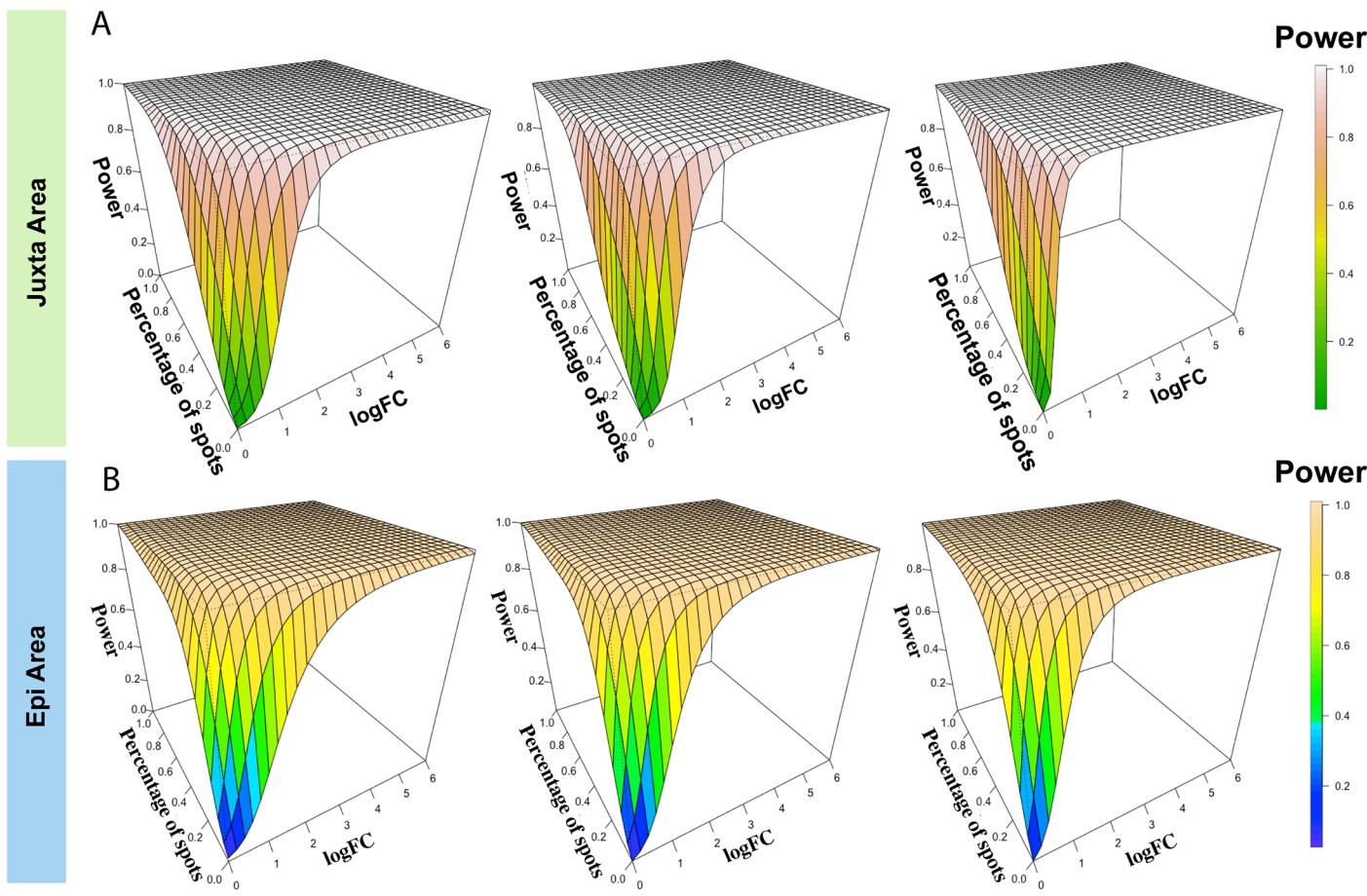

**Fig 3. Validation of the other two areas from the IPMN dataset.** The fitted power surfaces for slice replicate numbers of 6, 8, and 10 per group within (A) juxtalesional ('Juxta') areas and (B) epilesional ('Epi') areas.

0.04 for the epilesional and perilesional areas, respectively. These observations suggested that estimated power values for DEG detection across different functional regions in the IPMN samples are similar.

## 3.2. Power surface estimation with human colorectal cancer data

The second dataset we used for power analysis was a human colorectal cancer (CRC) dataset obtained by 10X Genomics [30]. The dataset contains 31 human colonic specimens with about 17,000 genes detected. From this dataset, we selected 7 tissue samples diagnosed as microsatellite instability-high (MSI-H) CRC and 6 tissue samples diagnosed as microsatellite stable (MSS) CRC, while holding out the remaining slices for independent validation. The sample keys for the slices included in the simulation and those for validation are listed in S2 and S3 Tables. As described by Heiser et al. [30], MSI-H CRCs are usually more immunogenic than their conventional MSS colorectal adenomas. Therefore, identifying the DEGs under the two disease subtypes is meaningful. As reviewed in the introduction section, existing methods focus on power estimation for NanoString GeoMx studies or on tasks such as detecting specific cell populations. None of these methods are directly applicable here. Therefore, we present the results using our proposed method and validate their accuracy from an empirical perspective.

**3.2.1. Power surface estimated using P-splines.** As shown in Fig 4B and 4C, two areas of CRC tissues (carcinoma and carcinoma border) were annotated by pathologists based on hematoxylin and eosin staining. We focused our analysis on the carcinoma border, which contained about 500 spots per slice. We resampled 500,1000,...,4500,5000 spots 100 times from the MSS and MSI-H samples to mimic the regional specimens of 1,2,...,9,10. The power estimation results are shown in Fig 4A. Because the number of spots within one slice in CRC samples was larger than that in the IPMN samples, a higher power value was achieved with the same parameter values. Crossings between power surfaces of different sample size values occurred only when the logFC small, where the power estimation may not have been numerically stable (Fig 4D). We further validated the power estimation results by randomly selecting a subset of slices from those used for simulation and comparing observed DE results to the predicted power. Specifically, with 4 slices per group, our method suggests a statistical power of 0.91 for detecting a DE gene with a detection rate of 0.05 and an absolute log-fold change of 0.3. We also performed the validation using independent slices that were initially held out. Both results (S4 and S5 Figs) confirm the power assessment of the proposed method.

**3.2.2. Local power estimation using XGBoost.** As a solution for crossing between power surfaces of different sample size values fitted by P-splines, XGBoost with 3D monotonic constraints can fit the power values locally. To prevent possible overfitting, we partitioned the dataset into 80% for training, 10% for validation, and 10% for testing and employed cross-validation and early stopping during model training. Additionally, we controlled model complexity by tuning hyperparameters, including maximum tree depth, number of parallel trees and learning rate based on the performance on the validation set (S6 Fig). The estimated power values for the logFC between 0.1 and 1 and expression rates between 0.05 and 0.15, as derived using XGBoost, are shown in Fig 5. These estimates resemble step functions with no intersections among surfaces of difference sample size. However, when we tried to implement XGBoost across a broader range of values for the logFC and expression rates, the power estimation proved ineffective (S7 Fig). This finding may be caused by the power values increasing rapidly with even minor increases in parameter values, which hindered the accurate power assessment via XGBoost's classification strategy. In contrast, quadratic spines implemented in

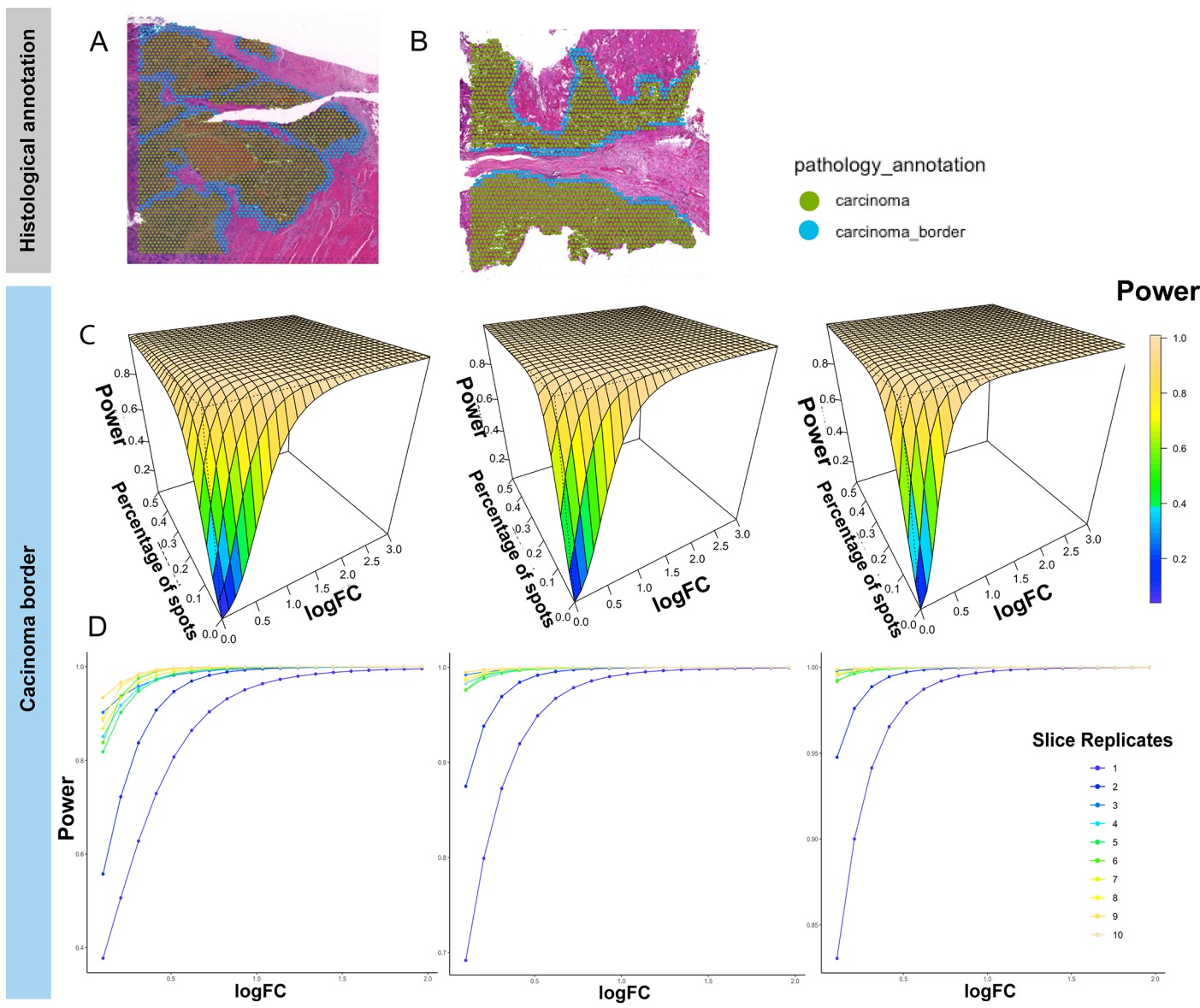

**Fig 4. Power estimation based on the CRC dataset.** (A and B) Histologically annotated carcinoma and carcinoma border areas in (A) an MSS sample and (B) an MSI-H sample. (C) The fitted power surfaces using P-splines for DE analysis within the carcinoma border with the number of slice replicates per group being 2, 4, and 6. (D) The relationship between the power and logFC in gene expression with slice replicates from 1 to 10 for the percentage of spots with the detected gene being 0.05, 0.10, and 0.15.

shape-constrained additive models are capable of catching such patterns. Therefore, XGBoost is recommended for local power value estimations when crossings occur in regions with parameters of specific interest.

Apart from XGBoost, we also implemented LightGBM [31], another machine learning model based on gradient boosting framework, with monotone constraints applied. Although LightGBM and XGBoost differ in their tree construction strategies (summarized in S4 Table), the results from LightGBM are consistent with those obtained from XGBoost (S8 and S9 Figs), indicating the robustness of our findings across gradient boosting implementations.

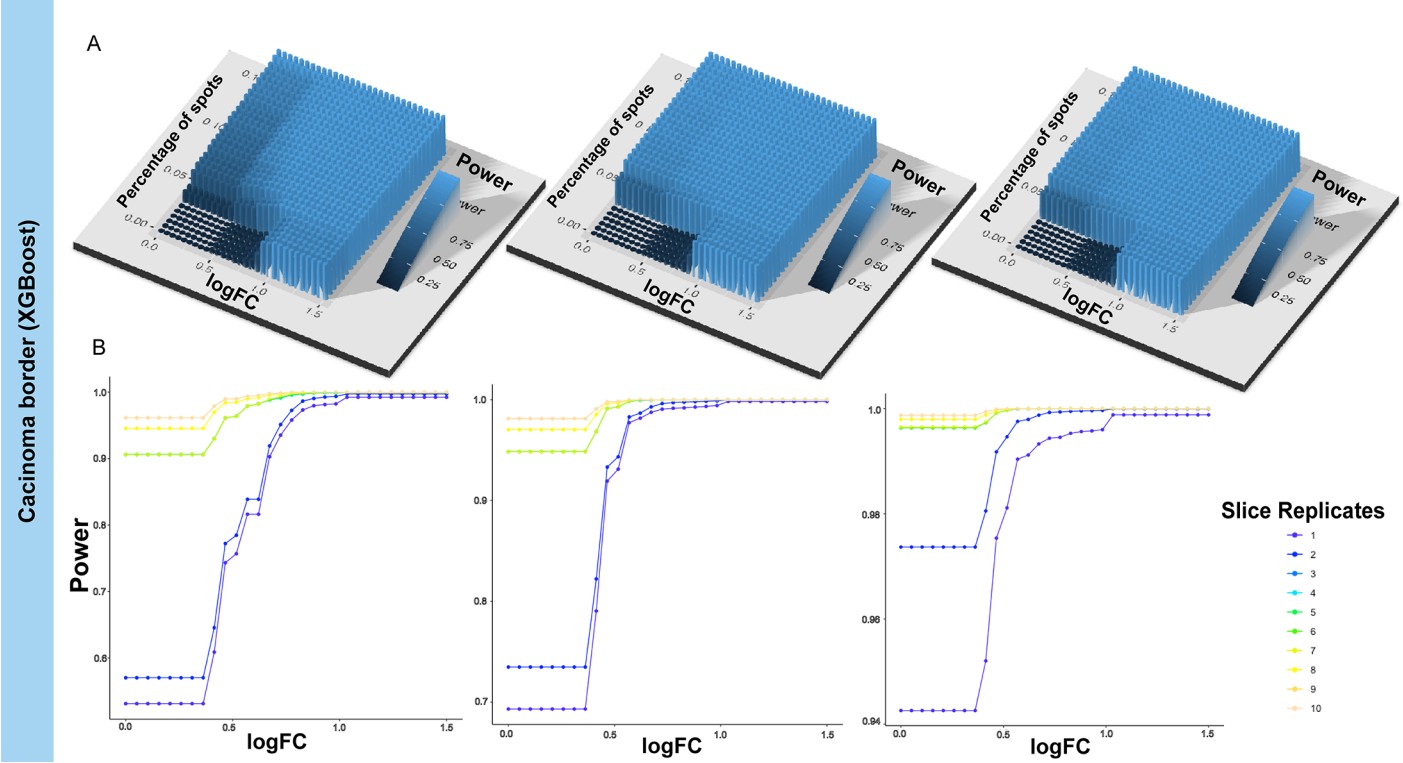

**Fig 5. Local power estimation based on the CRC dataset by XGBoost.** The power is fitted by XGBoost under constraints so that it monotonically increases with the percentage of expressed spots, the logFC and the number of slice replicates. The figures are based on power values where the logFC is between 0.1 and 1 and percentage of expressed spots is between 0.05 and 0.15. (A) The fitted power surfaces for DE analysis within the carcinoma border with the number of slice replicates per group being 2, 4, and 6. (B) The relationship between the power and logFC with slice replicates ranging from 1 to 10 for the percentage of spots with the detected gene being 0.05, 0.10, and 0.15.

## 4. Discussion

We developed PoweREST, a flexible method of power analysis, for detecting DEGs in two groups using ST data. One of the key considerations when designing a biomedical experiment is determining the appropriate sample size to ensure adequate statistical power. Whereas various methods have been developed for the power estimation using bulk RNA-seq and scRNA-seq, power estimation for ST data is underexplored owing to the challenges posed by its complex data structure and the integration of spatial information [5,14,15]. In the present study, we introduced a fully nonparametric pipeline to depict the power for experiments with 10X Genomics Visium ST data. Unlike methods for tunable *in silico* tissue generation based on parameterized models of tissue structure [13], our method used bootstrap resampling [24] to generate *in silico* ST samples. The estimated power values are then calculated by performing the DE analysis with the generated ST samples. Our method is fully nonparametric, which is also illustrated by using the shape-constrained P-splines [29] to fit the power surface along values of the logFC and the gene detection rate among spots. As an intrinsic requirement, 2D monotone constraints were imposed on the P-splines, which kept the monotonicity relationship between the estimated power values and the logFC and gene detection rate. XGBoost was proposed as a remedy to the crossings of power surfaces for different sample size values fitted by P-splines. To the best of our knowledge, PoweREST is so far the first study focusing on power estimation for detecting DEGs in 10X Visium ST experiments. As one of the few

studies attempting to perform power analysis for ST studies, it has the following advantages and limitations.

PoweREST incorporates the spatial information of ST data by computing test statistics using the spot-level gene expression within the same ROI. Rather than imposing a fixed spatial correlation structure, our nonparametric approach adapts to complex spatial relationships within the ROI, offering a more accurate representation of the power surface. This modeling approach assumes that the spatial correlation and DE gene distributions observed in the preliminary dataset are representative of that in future experiments. The ROI selection is typically guided by biological insights or tissue architecture with the help of biological investigators and pathologists. In this study, we focused on the carcinoma regions and their adjacent areas. However, alternative ROI selection criteria or study objectives may lead to different spatial characteristics, which in turn may influence the resulting power surface. In such cases, the estimated power may need to be adjusted to account for tissue heterogeneity, with the optimal sample size increasing in the presence of greater heterogeneity or decreasing when the target tissue is more homogeneous.

Compared with the existing method developed for NanoString GeoMx in the NAFLD fibrosis study [17], PoweREST offers both technical and numerical advantages in flexibility, accessibility, and power estimation performance. While the NAFLD framework is tailored to a specific application, staging liver fibrosis based on two selected genes (e.g., PON1 and FLNA), PoweREST provides a generalizable and non-parametric approach that supports any differential expression method, spatial structure, and allows user-defined input parameters such as gene detection rate and log-fold change. We summarized the detailed comparison in S5-S7 Tables. Though PoweREST is designed for 10X Genomics Visium platform and was evaluated on the Visium data in this study, its non-parametric framework is inherently flexible and can be extend to other ST platforms, as its simulation-based approach does not assume platform-specific parametric distributions. In future work, we plan to generalize PoweREST to support other ST platforms. Furthermore, our method is on spot level, thus incorporating cellular compositions may further improve the power estimation [32], which can also be our future research direction. When compared with existing approaches developed for bulk and single-cell RNA-seq data [16,33], our method uniquely incorporates spatial structure by bootstrapping spot-level gene expression within predefined ROIs, whereas prior methods either lack the resolution to address near-cellular measurements or assume independence between individual cells, limiting their applicability to spatial transcriptomics data (S8 Table).

PoweREST uses P-splines under 2D monotone constraints. Such constraints ensure the monotonically increasing relationships between power and logFC as well as between power and the gene detection rate. However, our method does not keep the monotonic relationship between power and sample size. Instead, the method fits the power surfaces separately for each value of sample size. In practice, we observed that the method keeps the monotonic relationship between estimated power and sample size in some cases. However, in other situations, this monotonic relationship is violated when the parameters approach 0. Currently, a robust software application for P-splines under 3D monotonic constraints is not readily available. To address this issue, we proposed using XGBoost, a machine learning technique that is capable of imposing three or more monotonic constraints on predictors. It essentially treats the prediction as a classification problem, which fits a visually stepwise function rather than a smooth curve [23]. However, the XGBoost method usually provides cruder estimates of the entire power surface than the P-spline method. Therefore, we recommend that users first use P-spline fitting and then evaluate the resulting fits. When crossings occur at the targeted

parameters, users should then use XGBoost but restrict its use to the regions where these crossings occur.

One potential disadvantage of our bootstrap-based approach is the heavier computational burden. In this report, the previous three steps were executed on a high-performance computing cluster using 12 CPU cores and a memory allocation of 56 GB. The bar plot (S10 Fig) shows the runtime versus the number of bootstraps used in an analysis, stratified by number of replicates per group. It was found that the run time is mainly determined by the times of bootstrap sampling rather than the replicates per group and at 100 bootstrap times, the runtime is about 78s. We also provided the pilot results of power estimations for DE analysis in the PoweREST R Shiny app for two different tissue types which takes less than 2s to generate the results. In the future, we aim to upload power estimation results for more cancer types. Additionally, in the PoweREST R package, we provided the option to prefilter genes based on their minimum detection rate and logFC to save computation time. We also included functions for power calculation focusing on genes specified by users. Our bootstrap sampling strategy relies on the assumption that batch effects have been adequately removed during preprocessing in earlier analysis. The datasets used in this study have already undergone such corrections by the data providers, ensuring that residual batch effects are minimal and unlikely to bias power estimation. Another limitation of a nonparametric method is that it may sometimes lead to relatively larger residuals when compared with parametric models. Such observations can be found in reports such as Dodd et al. [34], in which residuals from P-splines and those from a Poisson distribution were compared. Large residuals with P-splines may be the result of emphasis on smoothness at the expense of fit or the presence of noise in the data that the model smooths over. Therefore, residual plots should be checked to diagnose where the model may be underperforming and interpret those results carefully. The functions to create diagnosis plots are also included in the PoweREST R package.

## Conclusion

PoweREST is a nonparametric power estimation tool for spatial transcriptomics data used to detect differentially expressed genes under two conditions. Power results are influenced by the heterogeneous tissue structure, especially for cancer tissues, which can be captured by PoweREST but cannot be fully accounted for by parametric statistical models. It enables power calculation with and without prior spatial transcriptomics data available and is feasible for various differential expression analysis algorithms. It uses penalized splines under 2D-monotonic constraints to depict the power surface, which is biologically meaningful. We also provides a Shiny app with fitted power results for differential expression analysis of several carcinoma tissue types.

## Supporting information

**S1 Fig. Volcano plot of validation results upon IPMN dataset.**
(PDF)

**S2 Fig. The relationships between the estimated power and log fold change when the percentage of spots detecting the gene equals 0.1,0.2,0.3.** (A) Juxtalesional areas.
(B) Epilesional areas.
(PDF)

**S3 Fig. The relative difference between the estimated power surfaces.** The relative difference between the estimated power surfaces from perilesional areas and juxtalesional areas (A), and between the estimated power surfaces from perilesional areas and epilesional areas (B), when

the number of replicates per group is 6,8,10. The relative difference is calculated by comparing the difference between the estimated power values of two areas to the reference values. Specifically, it is computed using the Eq 2.

$$\text{Relative Difference} = \frac{|\text{Estimated Power (Juxta/Epi)} - \text{Estimated Power (Peri)}|}{\text{Estimated Power (Peri)}} \qquad (2)$$

(PDF)

**S4 Fig. Volcano plot of validation results upon CRC dataset.**
(PDF)

**S5 Fig. Volcano plot of validation results upon the independent CRC slices that were held out during model fitting.**
(PDF)

**S6 Fig. Tunning hyperparameters of XGBoost upon the validation set.** (A) Impact of Max Depth on Root Mean Square Error (RMSE) Across Different Learning Rates and Tree Counts. (B) Impact of Learning Rate on RMSE Across Different Max Depths and Tree Counts.
(PDF)

**S7 Fig. Estimation upon the entire power surface using XGBoost.** (A) The fitted power surfaces for DE analysis within the carcinoma border, under the slice replicates per group being 2,4,6. A top-down 2D view is provided above each corresponding 3D surface plot. (B) The relationship between the power and logFC under slice replicates from 1 to 10, for the percentage of spots detecting the gene being 0.05,0.1,0.15. Compare with Fig 5 in the main manuscript where XGBoost was used to fit power values where logFC between 0.1 and 1 and percentage of expressed spots between 0.05 and 0.15, the estimations here are less precise due to the characteristics of XGBoost's algorithm.
(PDF)

**S8 Fig. LightGBM results of fitted power values where the logFC is between 0.1 and 1 and percentage of expressed spots is between 0.05 and 0.15.** (A) The absolute difference values between the fitting results obtained from LightGBM and XGBoost. (B) The feature importance of the fitted models. (C) The fitted XGBoost model. (D) The fitted LightGBM model.
(PDF)

**S9 Fig. Comparison between fitted results of LightGBM and XGBoost.** (A) The absolute difference values between the fitting results obtained from LightGBM and XGBoost. (B) The feature importance of the fitted models. (C) The fitted XGBoost model. (D) The fitted Light-GBM model.
(PDF)

**S10 Fig. Runtime of PoweREST.** Computational time for PoweREST steps 1–3.
(PDF)

**S1 Table. Four differentially expressed genes with a detection rate around 0.1 and a log-fold change around 0.6.**
(PDF)

**S2 Table. Sample keys of slices that were included for simulation and model development.**
(PDF)

**S3 Table. Sample keys of slices that were held out for validation.**
(PDF)

**S4 Table. Comparison between XGBoost and LightGBM decision tree growth strategies and hyperparameters.**
(PDF)

**S5 Table. PoweREST vs. NAFLD Fibrosis Study Sample Size Design.**
(PDF)

**S6 Table. Final simulation results from NAFLD fibrosis sample size design.** (A) Primary endpoint PON1 in 165 $\mu m$ hepatocyte ROIs and 2 ROIs (2 Visium spots) per patient. Number of patients is indicated per group. (B) Secondary endpoint FLNA in 165 $\mu m$ in fibrotic niche and 2 ROIs (2 Visium spots) per patient. Number of patients is indicated per group.
(PDF)

**S7 Table. PoweREST's power estimations for IPMN's perilesional areas.** Estimated power values across varying log-fold changes and gene detection rates, stratified by number of slices per group (fixed at 80 spots per slice).
(PDF)

**S8 Table. PoweREST vs. Power estimation methods developed for bulk and single-cell RNA-seq data.**
(PDF)

**S1 Appendix. Intuition upon 3D monotonic relationships from the mathematical formula.**
(PDF)

**S2 Appendix. P-spline fitting under 2D monotonic constraints.**
(PDF)

## Acknowledgments

We thank Ashli Nguyen-Villarreal, Associate Scientific Editor, and Don Norwood, Scientific Editor, in the Research Medical Library at The University of Texas MD Anderson Cancer Center for editing this article.

## Author contributions

**Conceptualization:** Liang Li, Ziyi Li.

**Data curation:** Anirban Maitra, Ken Lau, Harsimran Kaur.

**Formal analysis:** Lan SHUI, Liang Li.

**Funding acquisition:** Ying Yuan.

**Investigation:** Lan SHUI, Liang Li, Ziyi Li.

**Methodology:** Ying Yuan, Liang Li, Ziyi Li.

**Software:** Lan SHUI.

**Supervision:** Liang Li, Ziyi Li.

**Validation:** Lan SHUI.

**Visualization:** Lan SHUI.

**Writing – original draft:** Lan SHUI, Liang Li, Ziyi Li.

**Writing – review & editing:** Lan SHUI, Anirban Maitra, Ying Yuan, Ken Lau, Harsimran Kaur, Liang Li, Ziyi Li.

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
