## [Decision Letter · Decision Letter 0]

14 Apr 2025

PCOMPBIOL-D-25-00403

PoweREST: Statistical Power Estimation for Spatial Transcriptomics Experiments to Detect Differentially Expressed Genes Between Two Conditions

PLOS Computational Biology

Dear Dr. SHUI,

Thank you for submitting your manuscript to PLOS Computational Biology. After careful consideration, we feel that it has merit but does not fully meet PLOS Computational Biology's publication criteria as it currently stands. Therefore, we invite you to submit a revised version of the manuscript that addresses the points raised during the review process.

Please submit your revised manuscript within 60 days Jun 14 2025 11:59PM. If you will need more time than this to complete your revisions, please reply to this message or contact the journal office at ploscompbiol@plos.org. Please include the following items when submitting your revised manuscript:

We look forward to receiving your revised manuscript.

Kind regards,

Joshua Welch

Academic Editor

PLOS Computational Biology

Jian Ma

Section Editor

PLOS Computational Biology

**Additional Editor Comments :**

The reviewers have raised important concerns about your work, all of which must be addressed before the paper can be accepted. In particular, it is crucial to clearly position your work against previous approaches for power estimation. In addition, both reviewers have raised key concerns about the technical details of your approach and its applicability to different types of spatial transcriptomic data.

**Journal Requirements:**

4) We notice that your supplementary Figures, and information are included in the manuscript file. Please remove them and upload them with the file type 'Supporting Information'. Please ensure that each Supporting Information file has a legend listed in the manuscript after the references list.

Potential Copyright Issues:

i) Figure 1. Thank you for stating that it is created with BioRender.com. Please confirm that you hold a Premium account and provide a pdf copy of the CC BY 4.0 License as provided by BioRender. For instructions on how to generate a CC BY 4.0 license for your figure, please see the guidelines here: https://help.biorender.com/hc/en-gb/articles/21282341238045-Publishing-in-open-access-resources.

If you are using the free assets from BioRender, we are unable to publish these images as they are licensed under a stricter license than CC BY 4.0. In this case we ask you to remove the BioRender images and replace them with open source alternatives.

See these open source resources you may use to replace images / clip-art:

- https://bioart.niaid.nih.gov/

- https://bioicons.com/

- https://healthicons.org/

- https://scidraw.io/

- https://reactome.org/icon-lib

- https://www.phylopic.org/images

ii) The following Figure contains a logo or branding: 1. We are not permitted to publish this under our CC-BY 4.0 license, even with permission. We ask that you please remove or replace it.

6) We note that your Data Availability Statement is currently as follows: "The authors are willing to make the data and computational code fully available." Please provide a complete Data Availability Statement in the submission form, ensuring you include all necessary access information or a reason for why you are unable to make your data freely accessible. If your research concerns only data provided within your submission, please write "All data are in the manuscript and/or supporting information files" as your Data Availability Statement. If your research concerns data from external sources, please amend your Data Availability Statement to include the full link to the data. Please note that authors must share the “minimal data set” for their submission. PLOS defines the minimal data set to consist of the data required to replicate all study findings reported in the article, as well as related metadata and methods (https://journals.plos.org/plosone/s/data-availability#loc-minimal-data-set-definition).

7) Please amend your detailed Financial Disclosure statement. This is published with the article. It must therefore be completed in full sentences and contain the exact wording you wish to be published.

**Reviewers' comments:**

Reviewer's Responses to Questions

Reviewer #1: The manuscript highlights the importance of power analysis in ST studies, which is often overlooked but crucial for designing robust experiments and interpreting results accurately. The development of PowerREST addresses a gap in the field of spatial transcriptomics (ST) by providing a tool for power estimation in detecting differentially expressed genes (DEGs) between two conditions. This is particularly important given the high cost and complexity of ST experiments. Here are some scientific comments on the manuscript:

1. The manuscript focuses exclusively on 10X Genomics Visium data, which uses fixed 55-µm spots. However, other ST platforms (e.g., Slide-seq, MERFISH, Seq-Scope, or NanoString GeoMx) have different spatial resolutions, data structures, and ROI selection methods. The tool's applicability to these platforms is not discussed.

2. The manuscript validates PowerREST using two publicly available datasets (IPMN and CRC). While this is a good start, the tool's performance on independent datasets or experimental validation (e.g., comparing predicted power with observed power in new experiments) is lacking.

3. The manuscript briefly mentions the NanoString GeoMx power estimation method but does not provide a detailed comparison with PowerREST. A more thorough comparison with existing methods (such as for single-cell, bulk RNA-seq data) would highlight the unique advantages and limitations of PowerREST.

4. While the manuscript demonstrates the application of PowerREST to IPMN and CRC datasets, it does not fully explore the biological or clinical implications of the findings.

5. While XGBoost is proposed as a solution for maintaining monotonicity, it is a machine learning model that can overfit the data, especially when applied to small or noisy datasets. This could lead to unreliable power estimates in some cases.

6. The manuscript demonstrates PowerREST on two cancer types (IPMN and CRC) but does not explore how tissue-specific characteristics (e.g., stromal content, immune infiltration) might influence power estimation. Different tissues may have varying levels of spatial heterogeneity, which could affect the accuracy of power estimates.

7. As we know, cancer tissues demonstrate the high levels of spatial heterogeneity. The power estimation highly depends on tissue slice selections or region selections.

8. PowerREST is primarily designed for power estimation in spatial transcriptomics, However, it is still unclear which parameter settings are most effective for capturing and leveraging spatial information. A more detailed discussion or analysis on how these parameters influence the power estimation, particularly in the context of spatial heterogeneity and tissue architecture, would be beneficial.

Reviewer #2: The authors proposed PoweREST, a nonparametric bootstrap-based tool for power analysis of detecting differentially expressed genes (DEGs) under two conditions in spatial transcriptomics (ST) data. More specifically, it is designed for ST data generated from 10X Genomics Visium technology. The method leverages penalized-spline to fit the powersurface, and it employs XGBoost for local estimation given crossings between power surfaces of different sample sizes. The authors provide both an R package and a R Shiny web app for users. My specific comments are listed below.

Major

1.The authors stated PoweREST implicitly incorporates the spatial information as it samples spots within the same ROI. The spatial information might be utilized for detecting spatial clusters or defining ROIs, which is not part of the PoweREST framework. Power analysis tool designed for scRNA-seq data is likely to be able to achieve the same purpose, provided with properly defined ROIs/spatial clusters.

2.Is PoweREST able to handle the situation where patient/donor-level difference exists (batch-effect, different number of replicates, imbalanced spot numbers in samples, etc) within each group? What would be the appropriate sampling strategy? Can PoweREST be used to calculate the minimum number of patients required to achieve desired statistical power in detecting DEGs?

3.Are there any other machine learning classification models able to impose multiple monotonic constraints on predictors? If so, it would be interesting to see their performance in comparison to XGBoost on local power estimation.

4.Given the spot-level data, the DEG results might reflect shifts in cellular composition rather than transcriptional change, it would be helpful to incorporate this information into the framework.

5.Can PoweREST be applied to high-resolution spatial transcriptomics data such as Xenium and CosMx?

Minor

1.In line 118, “ie” should be “i.e.”.

2.In Figure 5A, the axis labels are mismatched.

3.In line 289, the authors mentioned that they experimented with XGBoost using a larger range of logFC and detection rate. However, the result in Figure S3 comes from the

same range. The numbers of the pct axis in Figure S3A are hard to read.

4.The authors mentioned that the bootstrap-based method could be computationally expensive. What's the runtime for sampling and model fitting implemented on the data shown in the manuscript?

**Have the authors made all data and (if applicable) computational code underlying the findings in their manuscript fully available?**

Reviewer #1: Yes

Reviewer #2: Yes

PLOS authors have the option to publish the peer review history of their article (what does this mean?). If published, this will include your full peer review and any attached files.

Reviewer #1: No

Reviewer #2: No

**Figure resubmission:**
---

## [Decision Letter · Decision Letter 1]

2 Jul 2025

Dear MS SHUI,

We are pleased to inform you that your manuscript 'PoweREST: Statistical Power Estimation for Spatial Transcriptomics Experiments to Detect Differentially Expressed Genes Between Two Conditions' has been provisionally accepted for publication in PLOS Computational Biology.

Best regards,

Joshua Welch

Academic Editor

PLOS Computational Biology

Jian Ma

Section Editor

PLOS Computational Biology

Reviewer's Responses to Questions

**Comments to the Authors:**

Reviewer #1: All concerns raised have been addressed. No further comments in this round.

Reviewer #2: I appreciate the authors’ responses to the previous comments and their effort in performing additional analyses to clarify the concerns. I have no further comments.

**Have the authors made all data and (if applicable) computational code underlying the findings in their manuscript fully available?**

Reviewer #1: Yes

Reviewer #2: Yes

PLOS authors have the option to publish the peer review history of their article (what does this mean?). If published, this will include your full peer review and any attached files.

Reviewer #1: No

Reviewer #2: No

---

## [Editor Report · Acceptance letter]

PCOMPBIOL-D-25-00403R1

PoweREST: Statistical Power Estimation for Spatial Transcriptomics Experiments to Detect Differentially Expressed Genes Between Two Conditions

Dear Dr SHUI,

I am pleased to inform you that your manuscript has been formally accepted for publication in PLOS Computational Biology. Your manuscript is now with our production department and you will be notified of the publication date in due course.

With kind regards,

Anita Estes
